# Human Papillomavirus Infection and Vaccination: Knowledge, Attitude and Perception among Undergraduate Men and Women Healthcare University Students in Switzerland

**DOI:** 10.3390/vaccines7040130

**Published:** 2019-09-26

**Authors:** Emilien Jeannot, Manuela Viviano, Marie-Christine Follonier, Christelle Kaech, Nadine Oberhauser, Emmanuel Kabengele Mpinga, Pierre Vassilakos, Barbara Kaiser, Patrick Petignat

**Affiliations:** 1Faculty of Medicine, Institute of Global Health, Chemin de Mines 9, 1202 Geneva, Switzerland; Emmanuel.Kabengele@unige.ch; 2Community Psychiatric Service, Lausanne University Hospital (CHUV), 1011 Lausanne, Switzerland; 3Gynecology Division, Department of Obstetrics and Gynecology, Geneva University Hospitals, Boulevard de la Cluse 30, 1205 Geneva, Switzerland; manuela.viviano@hcuge.ch (M.V.); patrick.petignat@hcuge.ch (P.P.); 4School of Health Sciences (HESAV), 1011 Lausanne, Vaud, Switzerland; Marie-Christine.FOLLONIER@hesav.ch (M.-C.F.); christelle.kaech@hesav.ch (C.K.); nadine.oberhauser@hesav.ch (N.O.); 5Geneva Foundation for Medical Education and Research, Route de Ferney 150, 1211 Geneva, Switzerland; pierrevassilakos@bluewin.ch; 6University of Applied Sciences Western Switzerland, 2800 Délémont, Switzerland; barbara.kaiser@hes-so.ch

**Keywords:** cervical cancer, human papillomavirus (HPV), undergraduate students

## Abstract

Background: Human Papillomavirus is a common sexually transmitted infection, representing the main cause of genital warts and cervical cancer. The objective of this study was to evaluate basic knowledge and beliefs regarding HPV infection and HPV vaccine among undergraduate healthcare men and women students, as well as their attitudes towards HPV vaccine. Methods: Undergraduate women and men (nursing and midwifery curses) attending three Schools of Health Sciences located in Switzerland. A total of 427 women and 223 men have completed the web questionnaire, which included questions on their socio-demographic background and about basic knowledge and attitudes toward the HPV infection and vaccination. Results: Women undergraduate students have a better knowledge of HPV infection than their men counterparts, although there was a significant gap in knowledge of the disease’s mode of transmission and prevention. Among women, 72.6% of respondents reported having received at least one dose of HPV vaccines versus 31.4% for men respondents. Conclusion: The results of this study revealed a poor understanding among undergraduate healthcare men and women students about the HPV infection, its mode of transmission and its prevention. Our findings highlight the need to improve education on HPV for undergraduate healthcare students in order to increase the awareness of the disease.

## 1. Introduction

Human papillomavirus (HPV) is the most common viral of the reproductive system. Most sexually-active men and women will be infected with HPV at some point in their lives, while some of them will be repeatedly infected [1]. Although the majority of virus types are harmless, over 40 of them may cause cancer. Papilloma viruses can be transmitted through vaginal, oral, or anal sex. While they privilege the genital mucosae, these viruses can also reach the throat and cause pre-cancerous or cancerous lesions [2].

Two-thirds of HPV infections are asymptomatic. The persistence of high-risk HPV types, however, can cause various types of precancerous and cancerous lesions, including cervical cancer. In addition, HPV infections are responsible for other forms of cancer that can also affect men. Low-risk HPV can cause ano-genital warts (condyloma), which are common in both men and women. Over the course of life, 1 in 10 people on average will be affected [3].

In Switzerland, more than 5000 women are diagnosed with cervical pre-cancer each year and require further tests and/or surgery. They are most often young women, although cancer can sometimes only appear 20 or 30 years after the primary HPV Infection. Despite the implementation of screening, about 250 women will present cervical cancer in Switzerland every year [4].

Among over 200 HPV types, 14 of them can infect the genital organs in both women and men. Similarly to women, while infections may disappear over the years in some men, they may also persist in others. One recent study has shown that approximately three to four million cases of genital warts occur each year in men, with a peak rate of 500 per 100,000 in the 25–29 year-old men [5]. Another study conducted in the United States of America (USA) has estimated that about 2120 men in the U.S. will be diagnosed with cancer of the penis in 2017, while about 2950 men will be diagnosed with anal cancer [6].

In Switzerland, HPV infections were the main cause of anal cancer, which is diagnosed every year in 200 new cases, 90% of which are caused by HPV type 16 and 18 [7]. HPV can also take part in the development of other cancers in the genitals (penis) and throat. These cancers were, however, much rarer than those of the cervix and anus.

The introduction of the HPV vaccination represents the most important primary prevention measure against HPV-related precancer and cancer [8]. The currently available vaccines in Switzerland are Gardasil^®^ and Cervarix^®^, both of which protect against HPV genotypes 16 and 18. Gardasil^®^, which has been available on the international market since 2007, also covers against genotypes 6 and 11, which are mostly responsible for the development of genital condylomas [9]. This 9-valent vaccine, which protects against five additional types of oncogenic HPV (HPV 31, 33, 45, 52, 58), was launched on the Swiss market in 2016. The cantonal programs, however, have integrated it in their vaccination campaigns in 2019. If vaccination is successfully started before the 15th birthday, two injections at six-month intervals are recommended, starting from the 15th birthday, three injections over a period of at least six months are necessary for optimal protection. Swiss health authorities recommend vaccination against HPV to all teenagers aged 11 to 14 years. Since HPV-related diseases occur more frequently in women than in men, vaccination is recommended for girls as one of the mandatory vaccines, while for boys it is currently considered a supplementary vaccination. Since January 2016, Gardasil^®^ 9-valent was available and free of charge for boys in the majority of the Swiss cantons. The national coverage rates in Switzerland are assessed using the cantonal rates as part of the Swiss National Vaccination Coverage Survey (SNVCS). Concerning the HPV coverage rate, the most recent data are from the period 2014–2016. A study on the 2017–2019 period is currently underway, but the results are not yet available. For the period 2014–2016, the results were as follows: For two doses, after increasing from 24% in 2008–2010 to 54% in 2011–2013, the coverage rate no longer increased significantly during the period 2014–2016 and now stands at 56%. An analysis of the dose gap shows that during the last investigation period, only 48% of girls had received a valid two- or three-dose schema

One of the main challenges for the Swiss public health HPV vaccination program is to develop accurate forms of communication and information about the HPV infection. Research has identified that health professionals play an important role in vaccines uptake. Moreover, there is a lack of initiatives to improve education among undergraduate healthcare students about HPV infection, consequences, and prevention [10]. For a large majority of these young adults (women or men), the internet is the main and only source of information.

In order to make a conscious, informed decision about the vaccine’s uptake, the target population, which includes both women and men, should understand the importance of prevention through HPV vaccination, and the issues associated with the persistence of the infection.

Accurate knowledge about HPV infection and HPV vaccination are two critical points to make appropriate evidence-based health care choices. Consciousness about the knowledge of undergraduate health students on HPV infection and vaccination is important for the students themselves, but also for the society, as spreading the correct information about the vaccines is a fundamental point in ensuring community support [11].

Education of the community was, therefore, an essential step in the primary prevention of the HPV infection. This study aimed to evaluate (1) the basic knowledge and beliefs regarding HPV infections and HPV vaccines among undergraduate healthcare women and men students (nursing and midwifery) and (2) their attitudes towards the HPV vaccination.

## 2. Methods

### 2.1. Population

Recruitment of the study participants took place from January to March 2019 at three Schools of Health Sciences located in Switzerland. Men and women aged 18 years or older, currently attending these three Schools of Health Sciences to obtain a nursing or midwifery degree in their first year, second year, or third year, were invited to participate in the study.

### 2.2. Study Design

Announcements about the study were given by previously informed professors who were teaching classes at the School of Health Sciences. An email was also sent by the study investigators to the students prior to their recruitment.

The survey instrument was an online self-administrated anonymous questionnaire developed using SurveyMonkey software (Palo Alto, CA, USA). This software automatically saves responses into a secure database, thus protecting the participants’ confidentiality. On the first page of the web questionnaire, the participant could view a consent form, informing him/her of the study objectives and procedures. The participants had the right to refuse or terminate their participation in the study at any moment, in which case the time of study drop-out was indicated in the questionnaire. If the participants accepted to participate in the study, they were asked to tick a box in order to accept the informed consent form. If the participants did not agree to participate, the webpage automatically closed down. Three email reminders were sent at one, two weeks and three weeks after the first invitation, unless an individual requested to be removed from the mailing list throughout the process. The web-based survey was automatically closed 10 weeks after having sent the first invitation.

### 2.3. Study Tool

The questionnaire included three parts. The first part contained items about the socio-demographic characteristics of the participants. The second part contained items about basic knowledge of the HPV infection (17 items), and basic knowledge about HPV vaccination (seven items), where he/she could answer either “yes” or “no”. The third part contained items about the participants’ attitude toward the HPV vaccination (six items). The content’s validity was evaluated by three experts (nurse, midwife, and epidemiologist), and a feasibility study was previously performed on 15 nurses and five midwives (not publish). The questionnaire was developed in French, based on previous surveys evaluating HPV knowledge, attitudes and perceptions [12,13,14,15,16,17].

### 2.4. Sample Size

The total number of nursing and midwifery students enrolled at the three selected Schools of Health Sciences is around 1′200 students: A minimum sample size of *n* = 600 was calculated based on a confidence interval of 95%, a significance level of 0.05, a power of 80%, and response rate of 50%.

### 2.5. Statistical Analyses

Data collected by the Survey Monkey was exported to a Microsoft Excel database. Statistical analyses were run using STATA 13. Normality of the distribution was tested by the Kolmogorov–Smirnov test. Descriptive statistics and frequencies were analyzed for all variables. The *t*-test and Chi-square test were used for the descriptive statistics and for the comparison between variables. Logistic regression models were used to assess the associations between explicative variables and the status of the HPV vaccine’s uptake. The status of HPV immunization on women and men was used as the primary outcome. For this purpose, an individual was considered as vaccinated when he/she had received at least one dose of the vaccines. At the multivariate analysis, only those covariates considered to be of interest based on the univariate analysis’ results were included. All the hypotheses were two-sided, and results were considered significant at 0.05.

### 2.6. Ethical Approval

The study protocol was approved by the ethical cantonal board in Geneva (Commission Cantonale d’Ethique et de la Recherche—CCER) with the identification number Req-2019-00118. All participants signed an informed consent form prior to taking part in the study. The trial was registered under cliniclatrials.gov with the identifiers: NCT03888599.

## 3. Results

### 3.1. Participants Socio-Demographic Characteristics

A total of 650 men and women undergraduate students accepted to participate in the study and answered the entire questionnaire online and were thus included in the study.

The participants’ baseline characteristics are presented in Table 1. The mean of age was 23.1 years (range 18–35), 66% of the participants were nursing women students or midwives in their first year, second year or third year of Bachelor’s degree, the other 34% were nursing men students in either their first year, second year or third of Bachelor’s degree, while no midwifery man student took part in the study. The majority of the participants were Swiss (77%), 14% of them came from Europe (mainly France 10%) and 9% came from non-European countries (mainly either South America or Africa). The vast majority of the participants were not married (85%). A total of 65% of the participants were non-smokers. Overall, 9.2% of the women and men reported never having had sexual intercourse. The reported age of first intercourse of 17.5 years was the same for both women and men. A total of 14% of women students declared that they did not use a contraceptive method, 9.5% of the entire group had never used a condom, and 82% of them were sexually active.

### 3.2. Students Basic Knowledge about HPV

Table 2 shows the basic knowledge and beliefs about HPV infection and vaccination. Most of the students (women and men) knew that cervical cancer was strongly linked to the HPV infection (over 90% of positive responses), a vast majority of them was aware that HPV could to be sexually transmitted (86% of women and only 67% of men obtained positive response). The majority of women (84%) knew that HPV was responsible for genital warts, while only 61% of men answered this question. Overall, 75% of men and 42% of women believed that HPV infection could be treated with antibiotics. Nearly 50% of men students believed that men could not be infected with HPV.

### 3.3. HPV Vaccination’s Knowledge

The participants’ attitudes toward HPV vaccination are reported in Table 3. We observed that nearly 95% of women and 77% of men were aware of the existence of vaccines to protect women from HPV. Over 70.7% and 70.4% of women and men students, respectively, were aware of the existence of vaccines for both women and men. A total of 60.4% of women and 55.6% of men students believed that the HPV vaccine provided protection against most sexually transmitted infections.

### 3.4. Attitude toward the HPV Vaccines

The participants’ attitudes toward HPV vaccines are reported in Table 3. A total of 72.6% and 31.4% of women and men students, respectively, had received at least one dose of the vaccines. Overall, 29.5% of women and 42.2% of men students believed that the vaccination should be administered before the first sexual intercourse. Only 28.3% and 16.6% of the women and men students, respectively, knew that the vaccines were available for both women and men. A total of 89.5% of women and 90.1% of men students responded that they would recommend the HPV vaccination to their peers.

### 3.5. Predictors of HPV Vaccination

The results of the logistic regression predicting HPV vaccination are presented in Table 4.

Women participants were five times more likely to be vaccinated than their men counterparts (aOR: 5.79, 4.06–8.25 CI 95%). Participants with a European nationality also had higher vaccination rates than those with a Swiss nationality (aOR: 1.65, 1.42–1.92 CI 95%). Not being married, never having had sexual intercourse and not being sexually active at the moment were all predictive factors for having lower vaccination rates (aOR: 0.68, 0.22–0.72 CI 95%, aOR: 0.50, 0.30–0.83 CI 95% and aOR: 0.31, 0.10–0.95 CI 95%). A non-smoking status was also a predictor of a greater likelihood of being vaccinated for HPV (aor: 1.51, 1.05 - 2.81 CI 95%).

## 4. Discussion

This was the first study to assess knowledge about HPV infection and vaccination in a population of undergraduate men and women healthcare students in Switzerland. Previously published studies have sought to assess the prevalence of different HPV strains (only in nurses and midwife women) and the reasons and socio-demographic characteristics of the unvaccinated women [18,19]. This study also represents the first effort in evaluating the HPV vaccination coverage rate in a population of young men.

Our findings highlight a general lack of knowledge of the HPV infection’s natural history and its prevention among future Swiss nurses (men and women) and midwives (women only). The knowledge gaps of future health professionals have also been documented by studies conducted in other countries, such as Pakistan, Turkey, Lebanon, Germany, and USA [11,13,15,16,20].

Such knowledge gaps may be explained by the fact that among the three Schools of Health Sciences that participated in this study, none includes in their nursing curriculum a specific course about sexuality and HPV infection, with the exception of an optional course only available during their third year of bachelor. There is also no specific course about vaccinations in the nursing curriculum in these three schools. The topic of HPV is studied in the curriculum of midwives in a little more detail (about 2 h on their entire curriculum), two schools out of the three included in this study have a specific course (only 1 h) on vaccinations in their midwifery training curriculum.

This lack of education on HPV and other vaccines among future health professionals on the subject of vaccination (HPV, measles, and others) seems to be more frequent and is becoming a major problem due to the increasing hostility towards vaccination, particularly in the current context in Europe and the United States [21,22,23], where there is a growing mistrust towards vaccinations parallel to the increased incidence of vaccine-preventable diseases [24,25].

Our study indicates that young women have a higher level of knowledge about HPV than young men. This difference can be explained by the habit of girls to go for an annual check-up with the gynecologist or physician starting at puberty. Such consultations are aimed at providing girls with information about family planning, menstruation-related issues such as dysmenorrhea, and sexually transmitted diseases [26,27,28]. When asked about the source of their information about sexuality in general, young girls in Switzerland tend to turn to other girls, then, secondly, to magazines for young people and finally, to the Internet, while boys cite the Internet first and other young men as second [29]. While a study conducted in the United States found that the use of video messages was a potential tool to increase knowledge about HPV [30], other trials have also shown that social networks (e.g., Facebook, Instagram, Twitter etc.) can be used as complementary tools to deliver conventional prevention messages [31,32,33].

### Strength and Limitations of the Study

One of the strengths of our study is that it is the first in Switzerland to evaluate knowledge about HPV infection and HPV vaccination on such a large sample size. It is also the first to ask men about this problem in Switzerland and to have a first approach to HPV vaccination coverage for young men.

This study has some limitations that need to be addressed. The population sample was constituted of exclusively undergraduate students, which limits the generalization of our findings to the general population. As data were also collected through a questionnaire with self-reported answers, the reliability of which could not be directly verified by the study investigators, the results could also have been altered by such means of data collection.

HPV vaccination coverage rate was calculated using self-reporting of the number of doses received by participants to be sure of the number of doses people received, a copy of their vaccination carnet should have been requested, which was not possible in the context of this study. In the absence of a blood test, we cannot be sure of their HPV immunological status. It can, therefore, be assumed that the HPV vaccination coverage rate was calculated even if it was only a secondary objective of this study underestimates or on the contrary overestimates the right vaccination coverage rate. A final limitation of our study was the fact that the sample collected was not selected randomly, but according to the participation in the study by the students. This problem limits the generalization of our results.

## 5. Conclusions

The results of this study revealed a poor understanding among healthcare undergraduate men and women students about the HPV infection, its mode of transmission and its prevention. Nurses and midwives play a crucial role in shaping public views of HPV transmissions, prevention and vaccination. They represent a privileged channel to spread information about HPV to the target audience [34]. Future education campaigns and courses for healthcare students need to clarify multiple points about the modes of transmission of the infection, the means of prevention, including vaccination and other lesions induced by HPV in both men and women, in the view of increasing the vaccination coverage rate, and subsequently, reduce the rate of HPV-related cancers. In the future, those involved in prevention will have to make more use of the new communication channels in order to disseminate their message. Clear and targeted messages can positively influence adherence to primary and secondary preventive strategies, such as reduced risk-taking in sexual behavior or participation in HPV vaccination and cervical cancer screening.

The findings highlight the need for more HPV education among undergraduate healthcare student. It must be done to increase HPV knowledge and vaccination rates in this population.

## Figures and Tables

**Table 1 vaccines-07-00130-t001:** Socio-demographic characteristics of the study population.

	*N*	%
**Total**	650	
**Age (mean/SD)**	23.1	8.16
Range (min-max)	18–35	
**Gender**		
Women	427	65.7%
Men	223	34.3%
**Birthplace**		
Switzerland	502	77.2%
Europe	90	13.8%
Other	58	8.9%
**Relationship status**		
Married	554	85.2%
Not married	96	14.8%
**Smoker**		
Yes	423	65.1%
No	227	34.9%
**Ever had sexual Intercourse**		
Yes	559	86.0%
No	60	9.2%
Missing	31	4.8%
**Age of first sex encounter (mean/SD)**	17.5	1.83
**Number of sexual partners in lifetime (mean/SD)**	5.3	0.043
Women	3.6	7.8
Men	8.7	5.9
**Contraceptive method (question only for women *N* = 427)**		
vaginal ring	128	30.0%
hormonal IUD	16	3.7%
Injectable	2	0.5%
withdrawal	50	11.7%
condom	171	40.0%
no method	60	14.1%
**Condom use during sexual intercourse**		
Never	62	9.5%
Occasional	231	35.5%
Always	357	54.9%
**Currently sexually active**		
Yes	533	82.0%
No	117	18.0%

**Table 2 vaccines-07-00130-t002:** Knowledge and beliefs regarding HPV infections and HPV vaccines.

		Women *N* = 427				Men *N* = 223			
	Correct Answer	True Response		False Response		True Response		False Response	
HPV Knowledge Questions		*N*	%	*N*	%	*N*	%	*N*	%
The type of cancer highly associated with HPV infection is uterine cancer	True	400	93.7%	27	6.3%	205	91.9%	18	8.1%
HPV can be sexually transmitted	True	368	86.2%	59	13.8%	150	67.3%	73	32.7%
Having many sexual partners increases the risk of getting HPV	True	334	78.2%	93	21.8%	135	60.5%	88	39.5%
HPV can be passed on during sexual intercourse	True	298	69.8%	129	30.2%	125	56.1%	98	43.9%
A person could have HPV for many years without knowing it	True	267	62.5%	160	37.5%	135	60.5%	88	39.5%
HPV always has visible signs or symptoms	False	231	54.1%	196	45.9%	150	67.3%	73	32.7%
HPV is very rare infection	False	285	66.7%	142	33.3%	138	61.9%	85	38.1%
There are many types of HPV	True	306	71.7%	121	28.3%	147	65.9%	76	34.1%
Using condoms reduces the risk of getting HPV	True	370	86.7%	57	13.3%	187	83.9%	36	16.1%
HPV can be passed on by genital skin to skin contact	True	214	50.1%	213	49.9%	147	65.9%	76	34.1%
HPV can cause genital warts	True	360	84.3%	67	15.7%	136	61.0%	87	39.0%
HPV can cause herpes	False	258	60.4%	169	39.6%	141	63.2%	82	36.8%
HPV can be cured with antibiotics	False	245	57.4%	182	42.6%	54	24.2%	169	75.8%
Most sexually active people will get HPV at some point in their lives	True	201	47.1%	226	52.9%	157	70.4%	66	29.6%
Having sex at an early age increases the risk of getting HPV	True	220	51.5%	207	48.5%	109	48.9%	114	51.1%
HPV usually doesn’t need any treatment	True	235	55.0%	192	45.0%	104	46.6%	119	53.4%
Men cannot get HPV	False	350	82.0%	77	18.0%	120	53.8%	103	46.2%
**HPV Vaccine Knowledge Questions**									
There is a vaccine to protect women from HPV	True	405	94.8%	22	5.2%	173	77.6%	50	22.4%
There is a vaccine to protect men from HPV	True	302	70.7%	125	29.3%	157	70.4%	66	29.6%
The HPV vaccines offer protection against all sexually transmitted infections	False	258	60.4%	169	39.6%	124	55.6%	99	44.4%
Someone who has had HPV vaccine cannot develop cervical cancer	False	390	91.3%	37	8.7%	168	75.3%	55	24.7%
The HPV vaccines are most effective if given to people who have never had sex	True	367	85.9%	60	14.1%	162	72.6%	61	27.4%
The HPV vaccines offer protection against most cervical cancers	True	361	84.5%	66	15.5%	158	70.9%	65	29.1%
The HPV vaccine offers protection against genital warts	True	347	81.3%	80	18.7%	147	65.9%	76	34.1%

**Table 3 vaccines-07-00130-t003:** Attitudes toward HPV vaccines.

	Women *N* = 427		Men *N* = 223		*p*
	*N*	%	*N*	%	
**HPV vaccination status**					
Vaccinated (min 1 dose)	310	72.6%	70	31.4%	**<0.001 ***
Not vaccinated	117	27.4%	153	68.6%	
**Who should pay for this vaccination**					
Private Insurance	269	63.0%	126	56.5%	0.08
State health system	100	23.4%	61	27.4%	
From my pocket	23	5.4%	10	4.5%	
Others	10	2.3%	6	2.7%	
Don’t know	25	5.9%	20	9.0%	
**Who should get vaccinated**					
Women only	301	70.5%	185	83.0%	**0.0003**
Men and women	121	28.3%	37	16.6%	
Men only	5	1.2%	1	0.4%	
**When vaccine should be given**					
Before the first sex encounter	126	29.5%	94	42.2%	**0.0009**
Casual relationship	139	32.6%	67	30.0%	
If more than one partner	112	26.2%	43	19.3%	
Any time	50	11.7%	19	8.5%	
**Would you recommend the HPV vaccine?**					
Yes	382	89.5%	201	90.1%	0.69
No	45	10.5%	22	9.9%	
**Do you think that the vaccine should be offered free of charge**					
Yes	415	97.2%	220	98.7%	0.45
No	12	2.8%	3	1.3%	

* Bold font indicates a statistical significance.

**Table 4 vaccines-07-00130-t004:** Logistic regression predicting HPV vaccination (min one dose).

	aOR (95 CI)
**Gender**	
Men	Referent
Women	**5.79 (4.0 6–8.25)**
**Birthplace**	
Switzerland	Referent
Europe	**1.65 (1.42–1.92)**
Other	0.81 (0.66–1.03)
**Relationship status**	
Married	Referent
Not married	**0.68 (0.22–0.72)**
**Smoker**	
Yes	Referent
No	**1.51 (1.05–2.81)**
**Ever had sexual Intercourse**	
Yes	Referent
No	**0.50 (0.30–0.83)**
**Currently sexually active**	
Yes	Referent
No	**0.31 (0.10–0.95)**

Only odds ratio significant in the univariate model is presented in this table aOR adjusted OR for significant univariate predictors. Bold font indicates a statistical significance and a 95% confidence interval.

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
