# Peer review of "Human Papillomavirus Infection and Vaccination: Knowledge, Attitude and Perception among Undergraduate Men and Women Healthcare University Students in Switzerland"

_vaccines, 2019, doi:10.3390/vaccines7040130_

Round 1

Reviewer 1 Report

Comments to the Author;

Introduction:

There is no statistical data about HPV vaccines in Switzerland. So, authors should find references about vaccination against the virus in Switzerland. In the literature, several studies have focused on the attitudes and behaviours of at-risk groups and physicians about HPV infection and vaccination. The following references will help the authors.

1.Di Giuseppe, G.; Abbate, R.; Liguori, G.; Albano, L.; Angelillo, I.F. Human papillomavirus and vaccination: Knowledge, attitudes, and behavioural intention in adolescents and young women in Italy. Br. J. Cancer 2008, 99, 225–229.
2. Pelullo, C.P.; Di Giuseppe, G.; Angelillo, I.F. Human papillomavirus infection: Knowledge, attitudes, and behaviors among lesbian, gay men, and bisexual in Italy. PLoS ONE 2012, 7, e42856.
3. Bianco, A.; Pileggi, C.; Iozzo, F.; Nobile, C.G.; Pavia, M. Vaccination against human papilloma virus
infection in male adolescents: Knowledge, attitudes, and acceptability among parents in Italy. Hum. Vaccins Immunother. 2014, 10, 2536–2542.
4. Napolitano, F.; Napolitano, P.; Liguori, G.; Angelillo, I.F. Human papillomavirus infection and vaccination: Knowledge and attitudes among young males in Italy. Hum. Vaccins Immunother. 2016, 12, 1504–1510.
5. Napolitano, F.; Gualdieri, L.; Santagati, G.; Angelillo, I.F. Knowledge and attitudes toward HPV infection and vaccination among immigrants and refugees in Italy. Vaccine 2018, 36, 7536–7541.
6. Berkowitz, Z.; Malone, M.; Rodriguez, J.; Saraiya, M. Providers’ beliefs about the effectiveness of the HPV vaccine in preventing cancer and their recommended age groups for vaccination: Findings from a provider survey, 2012. Prev. Med. 2015, 81, 405–411.
7. Rohrbach, M.R.; Wieland, A.M. A survey of Wisconsin pediatricians’ knowledge and practices regarding the Human Papillomavirus vaccine. Otolaryngol. Head Neck Surg. 2017, 156, 636–641.
8. Bonville, C.A.; Domachowske, J.B.; Cibula, D.A.; Suryadevara, M. Immunization attitudes and practices among family medicine providers. Hum. Vaccins Immunother. 2017, 13, 2646–2653.
9. Caglioti, C.; Pileggi, C.; Nobile, C.G.; Pavia, M. Gynecologists and human papillomavirus DNA testing: Exploring knowledge, attitudes, and practice in Italy. Eur. J. Cancer Prev. 2017, 26, 249–256.
10. Napolitano, F.; Navaro, M.; Vezzosi, L.; Santagati, G.; Angelillo, I.F. Primary care pediatricians’ attitudes and practice towards HPV vaccination: A nationwide survey in Italy. PLoS ONE 2018, 13, e0194920.

Methodology:

I think the methodology too long. It is better to be shortened.

him/her page 7. I think no need for ‘his/her’,  the authors can use students/examiners instead of it.

Results:

I think the tables need to be rearranged, and their ligands need to be clearer. Tables in such a format are not accepted. There is a discrepancy in the values in the table (1) and the paragraph that appears above the table “56% of the participants were nursing female students or midwives”. So, the values need to be checked again. It is best to have the explanation on the tables after you have mentioned the table number directly in the text. 1st, 2nd or 3rd page 9 should be controled. Table 2 cannot be accepted as such; it needs to be rearranged.

There is repetition for discussion in page 18, I think it’s the conclusions.

Conclusions:

I think conclusions too long. It is better to be shortened. Future studies should address educational programs about HPV infection ….

Author Response

Comment 1:      There is no statistical data about HPV vaccines in Switzerland. So, authors should find references about vaccination against the virus in Switzerland”.

Answer 1:        Thank you for this comment, we have added a paragraph about HPV vaccines in Switzerland on introduction section

Comment 2:     “In the literature, several studies have focused on the attitudes and behaviours of at-risk groups and physicians about HPV infection and vaccination. The following references will help the authors”..

Answer 2:        Thank you for this complete reference list. We have added some of them to our article to improve its quality

Comment 3:     I think the methodology too long. It is better to be shortened”

Answer 3: Thank you for this remark but we prefer to leave the methodology part as complete as possible.

Comment 4:     “I think no need for ‘his/her’,  the authors can use students/examiners instead of it.

Answer 4: we have changed his/her for students

Comment 5:     “ I think the tables need to be rearranged, and their ligands need to be clearer. Tables in such a format are not accepted. There is a discrepancy in the values in the table (1) and the paragraph that appears above the table “56% of the participants were nursing female students or midwives”. So, the values need to be checked again. It is best to have the explanation on the tables after you have mentioned the table number directly in the text. 1st, 2nd or 3rd page 9 should be controled. Table 2 cannot be accepted as such; it needs to be rearranged”..

Answer 5:        Thank you for this comment, we have rearranged table for a better comprehension, we have also modified the discrepancy

Comment 6:     “ There is repetition for discussion in page 18, I think it’s the conclusions”..

Answer 6: exactly it’s a mistake. It’s not discussion but conclusions

Comment 7:     “ I think conclusions too long. It is better to be shortened. Future studies should address educational programs about HPV infection”..

Answer 7: Thank you for this remark but we prefer to leave the conclusion part as complete as possible

Reviewer 2 Report

Manuscript ID: vaccines-573519

Title: Human Papillomavirus infection and vaccination: knowledge, attitude

and perception among undergraduate men and women healthcare university

student in Switzerland

Overall, it is an interesting study.  The findings provide strong evidence for more education needed on HPV knowledge and vaccine for healthcare students.

I have several minor comments:

1.    The authors used “Male and female” and “men and women” interchangeably

2.    The Plural version would be more appropriated for “HPV infection”, and “vaccine”

3.    Page 3 the first sentence of the introduction “ Human papillomavirus (HPV) is the most common virus infection of the…” should be “the most common virus”

4.    Page 14 what does “predictors of HPV vaccine vaccination” mean?

5.    Page 17 the first sentence does not make sense.

6.    Page 18 “5 discussion”   Do you need two discussion sections?

7.    “Maried” in the tables should be “married”

Author Response

Comment 1:     « The authors used “Male and female” and “men and women” interchangeably

Answer 1 :        Thank you very much for your comment. We have changed in the text to keep consistency with the use of only "men and women".

Comment 2 :    “The Plural version would be more appropriated for “HPV infection”, and “vaccine. »

Answer 2 :        We have modified the text of the article to pluralize the terms of HPV infection and vaccines

Comment 3 :    « Page 3 the first sentence of the introduction “ Human papillomavirus (HPV) is the most common virus infection of the…” should be “the most common virus.»

Answer 3 :        Thank you for this comment, we have changed this sentence

Comment 4 :    .   « Page 14 what does “predictors of HPV vaccine vaccination” mean? »

Answer 4 : By this sentence we wanted to express our objective to know that what variable can influenced the fact of being vaccinated vs. not vaccinated. This was a secondary objective of our study to determine whether there were predictor’s variables/factors of being vaccinated or not

Comment 5:     « .    Page 17 the first sentence does not make sense.

Answer 5 :        We have changed this sentence.

Comment 6:     « .    Page 18 “5 discussion”   Do you need two discussion sections?.”

Answer 6 :        exactly it’s a mistake. It’s not discussion but conclusions

Comment 7:      .    “Maried” in the tables should be “married

Answer 7 :       thank you, we have change this mistake

Reviewer 3 Report

Jeannot et al., describe in their manuscript with the title: “Human Papillomavirus infection and vaccination: knowledge, attitude and perception among undergraduate men and women healthcare university students in Switzerland” present a survey to evaluate basic knowledge about HPV and the vaccination coverage among students in Switzerland.  Their results suggest that information and education on HPV has to be improved.

The study seems to be conducted and evaluated well. 

Yet, there are several mistakes and inconsistencies in the manuscript, which have to corrected.

Here in detail:

-in the introduction: The number of known HPV types, HPV types infecting the genital mucosa and HPV types associated with cancer is not correctly described and has to be carefully looked up. There are discrepancies, for instance on p. 3 is stated that “over 40 HPV types can cause cancer”, while p4 “Among over 150 HPV types, 40 of them can infect the genital organs”.  Not all HPV types infecting the genital mucosa are associated with cancer.  In addition, “HPV does not cause cancer”. Infection with HPV is associated with cancer” is the correct term.

-the numbers between in table 1 and the text are not conform. They should corrected

-Table 1: % should be removed in the heading lane.

-The headings of all tables should not appear within the table (see p11, p13, p14)

-P11: 86% of women knew that HPV is a sexual transmitted disease. This cannot be declared as little majority. Particularly, as in the following lane 84% is stated as majority. 

-The sentence “The participants attitude towards HPV vaccination is in table 3” should be deleted on p12. 

-P14: The term “Predictors of HPV vaccine vaccination” should corrected to “Predictors of HPV vaccination.

-P17: The sentence in the 2ndlane should be checked.

-P18: The term Discussion is used in 4. and 5.

Author Response

Comment 1:     « in the introduction: The number of known HPV types, HPV types infecting the genital mucosa and HPV types associated with cancer is not correctly described and has to be carefully looked up. There are discrepancies, for instance on p. 3 is stated that “over 40 HPV types can cause cancer”, while p4 “Among over 150 HPV types, 40 of them can infect the genital organs”.  Not all HPV types infecting the genital mucosa are associated with cancer.  In addition, “HPV does not cause cancer”. Infection with HPV is associated with cancer” is the correct term“.

»

Answer 1 :        Thank you very much for your comment we have modified this paragraph.

Comment 2:     “the numbers between in table 1 and the text are not conform. They should corrected)»

Answer 2 :        Thank you very much for your comment, we have modified the results section

. Comment 3:   “Table 1: % should be removed in the heading lane.

Answer 3 :       We do not agree with this comment and believe that the % should be kept in the table for a better understanding of the table 1

Comment 4:     “The headings of all tables should not appear within the table (see p11, p13, p14). “

Answer 4 :        exact, we have removed the heading of all tables

Comment 4:     “P11: 86% of women knew that HPV is a sexual transmitted disease. This cannot be declared as little majority. Particularly, as in the following lane 84% is stated as majority“. 

Answer 4 :       That’s true, we have modified this sentence

Comment 5:     “The sentence “The participants attitude towards HPV vaccination is in table 3” should be deleted on p12”.

Answer 5:        we have deleted this sentence

Comment 6:     “P14: The term “Predictors of HPV vaccine vaccination” should corrected to “Predictors of HPV vaccination“.

Answer 6:        We have changed this sentence

Comment 7:     “P17: The sentence in the 2ndlane should be checked “.

Answer 7:        Thank you for this comment. We have changed this paragraph

Comment 8:     “P18: The term Discussion is used in 4. and 5 “.

Answer 8 :       exactly it’s a mistake. It’s not discussion but conclusions

This manuscript is a resubmission of an earlier submission. The following is a list of the peer review reports and author responses from that submission.

Round 1

Reviewer 1 Report

This is a nice questionnaire study about knowledge of HPV and HPV vaccination in Switzerland. There are only minor aspects that I would recommend to be changed:

- Overall, there are several spelling mistakes and some passages that are difficult to understand because of poor English.

- In lines 69 and 70 the authors need to clarify that there are two different Gardasil vaccines: the original Gardasil from 2007 and the 9-valent Gardasil 9.

- In the results section please avoid double reporting in the text and the tables. Only the most importants Facts should be adressed in the text.

- Did really all students give written informed consent when they were approached by email?

- Please revise table 3. It is very confusing that the 3-digit-numbers are seperated.